
# Revisiting the flavor violating decays of the $\tau$ and $\mu$ leptons in the Standard Model with massive neutrinos

**Gerardo Hernández Tomé[1]⋆**

**1** Departamento de Física, Centro de Investigación y de Estudios Avanzados del Instituto Politécnico Nacional, Apdo. Postal 14-740, 07000 México D.F., México

⋆ ghernandez@fis.cinvestav.mx

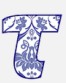

## Abstract

The flavor violating leptonic decays of the $\tau$ and $\mu$ leptons into three lighter charged leptons are revisited in the framework the Standard Model with massive neutrinos. In contrast to the previous prediction, we have found strongly suppressed rates for the $\tau^- \to \mu^- \ell^+ \ell^-$ ($\ell = \mu, e$) decays. Our results are in good agreement with the approximation of neglecting masses and momenta of the external particles in the loop integrals made in the first computation for the $\mu^- \to e^- e^+ e^-$ decay.

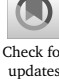

# 1 Introduction

The absence of right-handed neutrinos in the original formulation of the Standard Model (SM) implies massless neutrinos and lepton flavor conservation at any order in perturbation theory. Conversely, the discovery of neutrino oscillations [1] has demonstrated that lepton flavor numbers are not conserved in the neutrino sector and claims for an extended model with massive neutrinos.

In the simplest scenario of three light Dirac neutrinos, the mass matrix will be nondiagonal in the interaction (weak) basis, as occurs in the quark sector [2], and the mixing could be described through the $3 \times 3$ unitary Pontecorvo-Maki-Nakagawa-Sakata (PMNS) matrix [3]. Thus, charged lepton flavor violation (cLFV) transitions could arise at one loop level through charged flavor changing currents [1]. Nevertheless, it turns out natural to expect unobservable low rates, just as it has been reported for $BR(\ell^- \to \ell'^- \gamma) \sim \mathcal{O}(10^{-55})$ [4–6], $BR(Z \to \ell_i^- \ell_j^+) \sim \mathcal{O}(10^{-54})$ [7] and $BR(h \to \ell_i^- \ell_j^+) \sim \mathcal{O}(10^{-55})$ [8] which are far away from the capacity of any current or foreseen experimental facility.

In contrast, the prediction for the $\tau^- \to \mu^- \ell^+ \ell^-$ ($\ell = \mu, e$) decays given by ref. [9] report an unexpected value of $BR(\tau^- \to \mu^- \ell^+ \ell^-) \geq 10^{-14}$, but an updated evaluation using the expression for the amplitude derived in ref. [9] and employing the latest global fit results for neutrino mixing [10, 11] gives us a value of $BR(\tau^- \to \mu^- \ell^+ \ell^-) \sim 10^{-16}$. Furthermore, according to the results reported in [9], a value of $BR(\mu^- \to e^- e^+ e^-) \sim 10^{-21}$ would be predicted. This latter prediction disagree with an older computation for the $\mu^- \to e^- e^+ e^-$ decay in [5], where as a first approximation masses and momenta of the external particles are equal to zero.

Even though the updated predictions in [9] are still far away for the current experimental limits[2] (see for example Table 1) it is worth revisiting the two previous computation since there are at least some thirty orders of magnitude between both computations.

Table 1: Limits for the $\tau^- \to \mu^- \mu^+ \mu^-$, $\tau^- \to e^+ \mu^- \mu^-$ and $\tau^- \to e^- \mu^+ \mu^-$ decays set by the Belle, BaBar and LCHb collaborations. The last two columns stands for the projected sensitivity in Belle II and a tentative circular electron-positron collider. The values of this table have been extracted from [12].

| Decay channel | Belle ($10^{-8}$) | BaBar ($10^{-8}$) | LHCb ($10^{-8}$) | Belle II ($10^{-10}$) | FCC-ee ($10^{-12}$) |
|---|---|---|---|---|---|
| $\tau^- \to \mu^- \mu^+ \mu^-$ | 2.1 | 3.3 | 4.6 | 4.7-10 | 5-10 |
| $\tau^- \to e^+ \mu^- \mu^-$ | 1.7 | 2.6 | - | 3.6-4.7 | 5-10 |
| $\tau^- \to e^- \mu^+ \mu^-$ | 2.7 | 3.2 | - | 5.9-12 | 5-10 |

The $L^- \to \ell^- \ell'^- \ell'^+$ decays are induced through the diagrams depicted in fig. 1. Ref. [5] found that the dominant amplitudes are those with two neutrinos propagators[3], namely the penguin diagram (d) and the box diagram (e) in fig. 1. Conversely, the author in ref. [9] claims that only the penguin diagram (d) is relevant owing to the presence of a logarithmic divergent term depending on the neutrino mass.

As is well known, considering the effects or processes that arise from quantum corrections could involve divergent loop integrals. However, in any renormalizable theory, the possible divergences must vanish order by order (in the loop or effective field theory expansion) to be able to define (finite) observables. Furthermore, as neutrino oscillations, the LFV amplitudes

---

[1] There is still no evidence of cLFV, but strong constraints have been set in several channels. An extensive list of cLFV limits can be found in [4].

[2] The best limit for $BR(\mu^- \to e^- e^+ e^-) \leq 1.2 \cdot 10^{-11}$ was set by the SINDRUM experiment [10].

[3] In ref. [5] the amplitudes for diagrams (d) and (e) are proportional to $m_\nu^2 \log(m_\nu^2/m_W^2)$. Note that the presence of $m_\nu^2$ in the amplitude is responsible for the strong suppression rates.

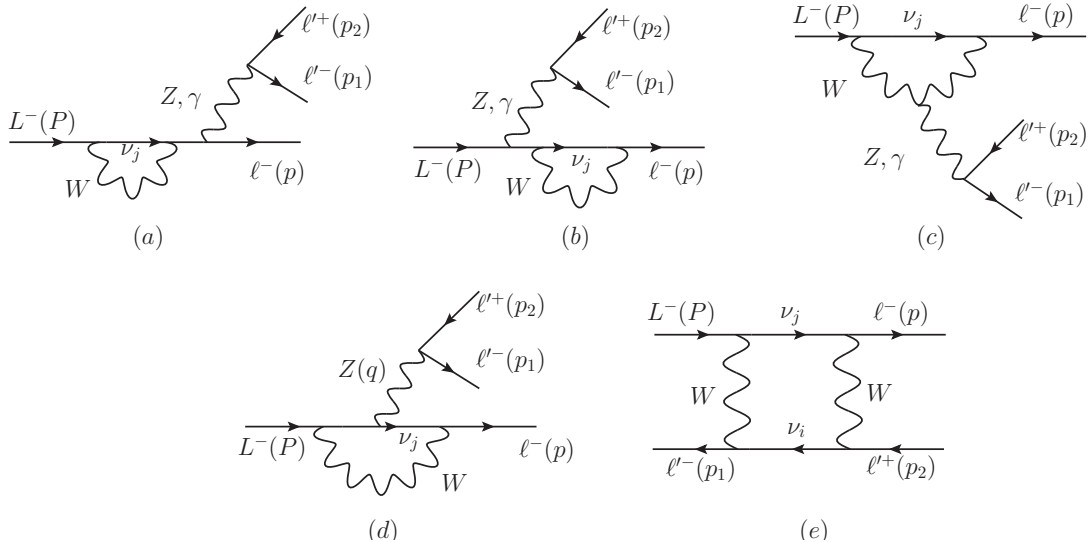

Figure 1: Feynman diagrams for the $L^- \rightarrow \ell^- \ell'^- \ell'^+$ decays in the presence of lepton mixing. Similar diagrams replacing the $W$ boson by the respective would-be Goldstone must be added in renormalizable $R_\xi$ gauges. Additionally, when $\ell = \ell'$ similar contributions (exchanging $p \leftrightarrow p_1$) to the amplitudes of diagrams (a) to (e) must be subtracted in order to antisymmetrize the amplitude. On the other hand, when $\ell \neq \ell'$, since the vertices of the neutral bosons $\gamma$ and $Z$ with a pair of fermions are flavor-conserving, only a similar (e) box diagram must be added interchanging $\ell(p) \leftrightarrow \ell'(p_1)$.

must vanish in the limit of degenerate neutrinos. Moreover, according to the Kinoshita-Lee-Nauenberg (KLN) theorem [13], the amplitude for massless neutrinos can go to zero, but it is impossible that it presents an IR divergence. This requirement is satisfied by the result of Ref. [5], but it is not the case in Ref. [9] which behaves as $\sum_j U_{Lj} U^*_{\ell j} \log(m_\nu/m_W)$ for very small neutrino masses.

The content of this work is the following, we first concentrate on the amplitude of the diagram (d) We show that the seeming logarithmic divergent behavior of the LFV amplitude reported in ref. [9] is not present, as the vanishing momentum transfer approximation considered in that paper lies outside the physical region. Then, in order to do a complete comparison with the computation in [5] we review the box contributions. Finally, we present our numerical results and conclusions.

## 2 Z-Penguin contribution emission from internal neutrino line

Following the convention used by the ref. [9] (see fig. 1) for masses and momenta of the external leptons, the amplitude of the diagram (d) can be written as

$$\mathcal{M}_d \quad \sim \quad \frac{i}{m_Z^2} l^\lambda_{L\ell} \times l_{\ell'\ell'\lambda}, \tag{1}$$

where $l_{\ell'\ell'\lambda} = -ig/(2c_W)\bar{u}_{p_1}\gamma_\lambda(g_v^{\ell'} - g_a^{\ell'}\gamma_5)v_{p_2}$ [4] is independent of the loop integration, whereas the relevant effective $ZL\ell$ transition is given as follows:

$$l_{L\ell}^\lambda = \left(\frac{-ig}{4c_W}\right)\left(\frac{-ig}{2\sqrt{2}}\right)^2 \sum_{j=1}^3 U_{\ell j}^* U_{Lj}\bar{u}_p \Gamma_j^\lambda u_P, \tag{2}$$

where $U_{im}$ are entries of the PMNS mixing matrix. In the Feynman-'t Hooft gauge, we have

$$\Gamma_j^\lambda = \int \frac{d^4k}{(2\pi)^4} \frac{\gamma_\rho(1-\gamma_5)i\left[(\slashed{p}+\slashed{k})+m_j\right]\gamma^\lambda(1-\gamma_5)i\left[(\slashed{P}+\slashed{k})+m_j\right]\gamma_\sigma(1-\gamma_5)(-ig^{\rho\sigma})}{\left[(p+k)^2-m_j^2\right]\left[(P+k)^2-m_j^2\right]\left[k^2-m_W^2\right]}. \tag{3}$$

After making the loop integration using dimensional regularization in order to deal with the (logarithmic) UV divergences, the Lorentz structure for the $\Gamma_j^\lambda$ factor can be written as follows,

$$\begin{aligned}
\Gamma_j^\lambda &= F_a\gamma^\lambda(1-\gamma_5) + F_b\gamma^\lambda(1+\gamma_5) + F_c(P+p)^\lambda(1+\gamma_5) \\
&+ F_d(P+p)^\lambda(1-\gamma_5) + F_e q^\lambda(1+\gamma_5) + F_f q^\lambda(1-\gamma_5),
\end{aligned} \tag{4}$$

where in general $F_k = F_k(q^2, m_j^2)$ ($k = a, b..., f$) with $q^\mu = (P-p)^\mu$ the momentum transfer by the $Z$ boson, and $m_j$ the mass of the neutrino. Of course $F_k$ functions will also depend on the mass of the $W$ gauge boson and external masses, but these have well-defined values.

Neglecting the momenta of the external particles in eq. (3) simplifies considerably the computation, as the only possible contribution is given by the $F_a^0$ function, where we are using a superscript 0 in order to distinguish this approximation. In this simple case, the $F_a^0$ function will not depend on $q^2$ and the integrals turn easily to solve analytically using either Feynman parameters or Passarino-Veltman method. In such a way that after making an expansion around $m_j^2 = 0$ we obtained

$$F_a^0 = \frac{1}{2\pi^2}\left[\frac{m_j^2}{m_W^2}\log\left(\frac{m_W^2}{m_j^2}\right) - \frac{m_j^2}{2m_W^2} + \frac{1}{2}\log\left(\frac{m_W^2}{\mu^2}\right) + \frac{1}{4} + \vartheta\left(\frac{m_j^2}{m_W^2}\right)^2\right]. \tag{5}$$

From eq. (5) it turns clear that the amplitude is proportional to the neutrino mass squared, and the dominant contribution, due to the big gap between the neutrino and $W$ boson mass scales, comes from the first term as it involves a relative factor $\log\left(\frac{m_W^2}{m_j^2}\right)$ compared to the second one, whereas the independent terms of neutrino masses will vanish by a GIM-like mechanism. Therefore, the structure of the matrix element for the contribution of the diagram (d) in fig. 1 is given by

$$\begin{aligned}
\mathcal{M}_d^0 &= -i\frac{G_F^2 m_W^2 \beta_{F_a^0}}{4}\bar{u}_p\gamma_\lambda(1-\gamma_5)u_P \times \bar{u}_{p_1}\gamma^\lambda(1-\gamma_5)v_{p_2} \\
&+ iG_F^2 m_W^2 s_W^2 \beta_{F_a^0}\bar{u}_p\gamma_\lambda(1-\gamma_5)u(P) \times \bar{u}_{p_1}\gamma^\lambda v_{p_2},
\end{aligned} \tag{6}$$

---

[4] $g$ is the $SU(2)_L$ coupling and $c_W$ ($s_W$) is short for the cosine(sine) of the weak mixing angle $\theta_W$. In the SM, $g_v^{\ell'} = -1/2 + 2s_W^2$ and $g_a^{\ell'} = -1/2$.

where we have defined

$$\beta_{F_a^0} = \sum_j U_{Lj} U_{\ell j}^* F_a^0(m_j^2). \tag{7}$$

Eq. (6) reproduces the result reported in ref. [5] considering only the first term in eq. (5) and the simple case of two families.

Returning to the general case (non-zero masses and momenta of the external particles), we obtained the $F_k$ functions using both Feynman parametrization and the Passarino-Veltman (PaVe) technique denoted by $F_{F_k}$ and $F_{PV_k}$, respectively. We agree with the expressions previously reported in ref. [9] in terms of the Feynman parameters [5], namely the $F_{F_k}$ functions can be written as

$$F_{F_k}(q^2, m_j^2) = \frac{1}{2\pi^2} \int_0^1 dx \int_0^{1-x} f_k(q^2, m_j^2) dy, \tag{8}$$

where

$$f_a = 2 + \log\left(D_j(q^2)/\mu^2\right) + \frac{(q^2 - m^2)x(y-1) + M^2 x(x+y) + q^2 y(y-1)}{D_j}, \tag{9}$$

$$f_b = \frac{mMx}{D_j}, \tag{10}$$

$$f_c = -\frac{Mx(x+y)}{D_j}, \tag{11}$$

$$f_d = -\frac{mx(1-y)}{D_j}, \tag{12}$$

$$f_e = \frac{Mx(2-3y-x) - 2My(y-1)}{D_j}, \tag{13}$$

$$f_f = \frac{xm(y-1) + 2my(y-1)}{D_j}, \tag{14}$$

and $D_j$ is defined as

$$D_j(q^2, m_j^2) = -(x-1)m_j^2 - m^2 xy + xm_W^2 + M^2 x(x+y-1) - q^2 y(1-x-y). \tag{15}$$

We have omitted in $f_a$ the term associated with the UV divergence since it is independent of $m_j$ and vanishes owing to the GIM-like mechanism.

In terms of the PaVe scalar functions the $F_k$ functions are given as follows

$$F_{PV_k}(q^2, m_j^2) = \frac{1}{2\pi^2} \frac{N_{F_k}}{D_{F_k}}, \tag{16}$$

with

$$D_{F_a} = 2D_{F_b} = -2\lambda(m^2, M^2, q^2), \quad D_{F_c} = D_{F_e} = \frac{M}{2} D_{F_a}^2 \quad D_{F_d} = D_{F_f} = \frac{m}{2} D_{F_a}^2, \tag{17}$$

---

[5]We have found some irrelevant differences in the numerators of the $f_d$ and $f_f$ functions, as can be seen comparing eqs. (12) and (14) with the corresponding expressions in ref. [9].

$$
\begin{aligned}
N_{F_k} &= \xi_{k_1} B_0(m^2, m_j^2, m_W^2) + \xi_{k_2} B_0(M^2, m_j^2, m_W^2) + \xi_{k_3} B_0(q^2, m_j^2, m_j^2) \\
&+ \xi_{k_4} B_0(0, m_j^2, m_W^2) + \xi_{k_5} C_0(m^2, M^2, q^2, m_j^2, m_W^2, m_j^2) + \xi_{k_0},
\end{aligned} \tag{18}
$$

where $\lambda$ is the Kallen function $\lambda(x, y, z) = x^2 + y^2 + z^2 - 2(xy + xz + yz)$, and in order to avoid lengthy expressions the $\xi_k$ factors can be found in [14].

Unlike the approximation made in ref. [5], the presence of masses and momenta of the external particles in the computation complicates the way for the derivation of analytical expressions for the integrals in eqs. (8) or (16). Nevertheless, in order to verify the equality between both expressions we have done a numerical cross-check, where we have employed the Looptools package [15,16] for the evaluation of the PaVe functions and a numerical Mathematica [17] routine for the evaluation of the parametric integrals.

At this point, we want to stress that we disagree with the approximation in ref. [9], where an expansion around $q^2 = 0$ is made in eq. (9) in order to estimate the relevant dependence on the neutrino mass of the $F_a$ function. We highlight that the dependence on $q^2$ plays a crucial role in the behavior of the $F_k$ functions. Moreover, we are studying a process where $q^2$ must be non-vanishing and is indeed much larger than the neutrino squared mass. Then, taking such expansion modifies substantially the behavior of the original functions in the interesting physical region for the neutrino masses and, as a consequence, it gives rise to an incorrect infrared logarithmically divergent behavior of the $F_k$ functions when $m_j$ goes to zero, without any possible cure. We point out the presence of a small imaginary part in the $F_a$ function, which emerges for the physical values $4m_j^2 < q^2$.

The $q^2$ minimum in the $L^- \to \ell^- \ell'^- \ell'^+$ decay is given by $4m_{\ell'}^2$, which is much larger than neutrinos masses. This, together with the difficulties in obtaining analytical expressions directly for the $F_k$ functions suggests employing some numerical approximation to deal with the problem. Because of this, we approximate the $F_k$ functions in the physical region for the neutrinos masses by fitting the curves for the real and imaginary parts of the $F_k$ functions evaluated in terms of the PaVe function. We have found a reasonably good fit of the form

$$
F_k = \frac{1}{2\pi^2 u} \left( Q_k + \frac{m_j^2}{m_W^2} R_k \right),
$$

where $u = 1$ for $k = a, b$ and $u = M$ for $k = c, d, e, f$ and tables with the respective values for the $Q_k = Q_{R_k} + iQ_{R_I}$ and $R_k = R_{R_k} + iR_{R_I}$ factors of all considered channels are given in [14]. It is clear that the $Q_k$ factors will not contribute owing to the GIM-like mechanism, whereas the relevant contributions are given by the $R_k$ factors. According to our numerical results, we find that the $R_k$ factors of the $F_b$, $F_c$ and $F_d$ functions are suppressed with respect to the $F_a$ factor. On the other hand, despite the respective factors of $F_e$ and $F_f$ functions are larger than those of the $F_a$ function, when the momentum transfer becomes smaller and smaller their helicity suppression makes them negligible. Thus, we will concentrate on the contribution of the $F_a$ function.

In order to check our results, we also have made an expansion for the PaVe functions involved in eq. (18), following the same strategy that Cheng and Li for the $\mu \to e\gamma$ decay [6], that is: expanding the loop integrals around $m_j^2 = 0$. It must be noted that, since neutrino masses are the smallest energy scale in the problem, this is the expansion that is most efficient for the considered decays. Using the Package-X program [18], we could rewrite the $F_{PV_a}$ contribution as follows:

$$
F_{PV_a}(q^2, m_j^2) = \frac{1}{2\pi^2} \left[ Q_a + \frac{m_j^2}{m_W^2} R_a + \vartheta\left( \frac{m_j^4}{m_W^4} \right) \right], \tag{19}
$$

where

$$Q_a = -\lambda(m^2, M^2, q^2)^{-1}\left[f_{Q_{a_1}} C_0(m^2, M^2, q^2, 0, m_W^2, 0) + f_{Q_{a_2}} \log\left(\frac{m_W^2}{m_W^2 - m^2}\right)\right.$$
$$+ \left. f_{Q_{a_3}} \log\left(\frac{m_W^2}{m_W^2 - M^2}\right) + f_{Q_{a_4}} \log\left(\frac{m_W^2}{q^2}\right) + f_{Q_{a_5}}\right] - \frac{1}{2}\Delta, \tag{20}$$

$$R_a = -m_W^2\lambda(m^2, M^2, q^2)^{-1}\left[f_{R_{a_1}} C_0(m^2, M^2, q^2, 0, m_W^2, 0) + f_{R_{a_2}} \log\left(\frac{m_W^2}{m_W^2 - m^2}\right)\right.$$
$$+ \left. f_{R_{a_3}} \log\left(\frac{m_W^2}{m_W^2 - M^2}\right) + f_{R_{a_4}} \log\left(\frac{m_W^2}{q^2}\right) + f_{R_{a_5}}\right], \tag{21}$$

in which $\Delta = \frac{1}{\epsilon} - \gamma_E + \log(4\pi)$, and the $f_Q$ and $f_R$ factors can be found in [14]. We consider the results obtained from eq. (21) for the effective vertices as our reference ones.

Finally, we can approximate the amplitude for the diagram (d) according to eq. (6) replacing $F_a^0$ by

$$F_a \approx \frac{1}{2\pi^2}\frac{m_j^2}{m_W^2}R_a. \tag{22}$$

## 3 Contributions of the box diagrams

Unlike the penguin diagram (d), which involves two neutrino propagators of the same mass state, the box diagram (e) can involve two neutrino propagators with different mass states. Thus, in full generality, the amplitude can be written as follows

$$\mathcal{M}_e = \left(\frac{-ig}{2\sqrt{2}}\right)^4 \sum_{i,j} U_{Lj}U_{lj}^*U_{\ell'i}U_{\ell'i}^* T_{\sigma\sigma'}I^{\sigma\sigma'}, \tag{23}$$

where we defined

$$T_{\sigma\sigma'} = 4\bar{u}_p\gamma_\mu\gamma_\sigma\gamma_\nu(1 - \gamma_5)u_P \times \bar{u}_{p_1}\gamma^\nu\gamma_{\sigma'}\gamma^\mu(1 - \gamma_5)v_{p_2}, \tag{24}$$

and the relevant loop integral is given by (see fig. 1 (e))

$$I^{\sigma\sigma'} = \int \frac{d^4k}{(2\pi)^4}\frac{(P + k)^\sigma(k + p_1)^{\sigma'}}{(k^2 - m_W^2)[(p_1 + p_2 + k)^2 - m_W^2][(P + k)^2 - m_j^2][(k + p_1)^2 - m_i^2]}. \tag{25}$$

Since we have written the eq. (25) in terms of the momenta $P$, $p_1$ and $p_2$, the integral must take the form

$$I^{\sigma\sigma'} = i\left(g^{\sigma\sigma'}H_a + P^\sigma P^{\sigma'}H_b + P^\sigma p_1^{\sigma'}H_c + P^\sigma p_2^{\sigma'}H_d + p_1^\sigma P^{\sigma'}H_e + p_1^\sigma p_1^{\sigma'}H_f\right.$$
$$+ \left. p_1^\sigma p_2^{\sigma'}H_g + p_2^\sigma P^{\sigma'}H_h + p_2^\sigma p_1^{\sigma'}H_i + p_2^\sigma p_2^{\sigma'}H_j\right). \tag{26}$$

In general $H_k = H_k(s_{12}, s_{13}, m_i^2, m_j^3)$, where $s_{12} = (p_1 + p_2)^2 = q^2$, $s_{13} = (p_1 + p)^2$. Again, in the approximation where momenta of the external particles are neglected in eq. (25), the things

go easily, since the only contribution is given by the $H_a^0$ function, which will not depend either on $s_{12}$ or $s_{13}$. In this case, after solving analytically the loop integrals and making a double Taylor expansion, first around $m_i^2 = 0$ and then around $m_j^2 = 0$, we obtained that

$$
\begin{aligned}
H_a^0(m_i^2, m_j^2) &= \frac{1}{64\pi^2 m_W^4}\left[\left(m_i^2 + m_j^2\right)\left(\log\left(\frac{m_W^2}{m_j^2}\right) - 1\right) + \frac{m_i^2 m_j^2}{m_W^2}\left(2\log\left(\frac{m_W^2}{m_j^2}\right) - 1\right)\right. \\
&\quad \left. - m_W^2 + \vartheta\left(\frac{m_i^4}{m_W^2}\right) + \vartheta\left(\frac{m_j^4}{m_W^2}\right)\right].
\end{aligned}
\tag{27}
$$

Using that $T_{\sigma\sigma'}g^{\sigma\sigma'} = 16\bar{u}_p\gamma_\lambda(1-\gamma_5)u_P \times \bar{u}_{p_1}\gamma^\lambda(1-\gamma_5)v_{p_2}$, the amplitude in this approximation is given by

$$
\mathcal{M}_e^0 = i8G_F^2 m_W^4 \beta_{H_a^0} \bar{u}_p\gamma_\lambda(1-\gamma_5)u_P \times \bar{u}_{p_1}\gamma^\lambda(1-\gamma_5)v_{p_2},
\tag{28}
$$

with

$$
\beta_{H_a^0} = \sum_{j,i} U_{Lj}U_{\ell j}^* U_{\ell' i}U_{\ell' i}^* H_a^0(m_i^2, m_j^2).
\tag{29}
$$

Taking the first term in eq. (27) and considering only two families, the eq. (28) reproduces the expression reported in ref. [5]. Furthermore, this result is consistent with the previous expression reported in Ref. [19] for the box contribution associated with the effective $K^+ \to \pi^+ \nu_\ell \bar{\nu}_\ell$ decay in the quark sector, where the approximation of taking masses and momenta of the external particles equal to zero is excellent, owing to the presence of the heavy top quark inside the loop.

In the general case, we obtained the $H_k$ ($k = a, b, ..., j$) functions in terms of both Feynman parameters integrals, $H_{F_k}$, and PaVe functions, $H_{PV_k}$. Using Feynman parametrization these functions read

$$
H_{F_k}(s_{12}, s_{13}, m_i^2, m_j^2) = \frac{1}{16\pi^2}\int_0^1 dx \int_0^{1-x} dy \int_0^{1-x-y} h_k\, dz,
\tag{30}
$$

where

$$
h_a = -\frac{1}{2M_F^2}, \quad h_b = \frac{z(z-1)}{M_F^4}, \quad h_c = -\frac{(z-1)(x+z)}{M_F^4}, \quad h_d = \frac{y(z-1)}{M_F^4},
\tag{31}
$$

$$
h_e = -\frac{z(x+z-1)}{M_F^4}, \quad h_f = \frac{(x+z-1)(x+z)}{M_F^4}, \quad h_g = -\frac{y(x+z-1)}{M_F^4},
\tag{32}
$$

$$
h_h = \frac{yz}{M_F^4} \quad h_i = -\frac{y(x+z)}{M_F^4}, \quad h_j = \frac{y^2}{M_F^4},
\tag{33}
$$

where we have defined $M_F^2$ as follows

$$
\begin{aligned}
M_F^2 &= -m_j^2(x+y-1) + m_{\ell'}^2(x+y-1)(x+y) + m_W^2(x+y) - s_{12}xy \\
&\quad + z^2\left(2m_{\ell'}^2 + m^2 + M^2 - s_{12} - s_{13}\right) + z\left[m_i^2 - m_j^2 + (x+y)\left(3m_{\ell'}^2 - s_{12} - s_{13}\right) - 2m_{\ell'}^2\right. \\
&\quad \left. + m^2(x-1) + M^2(y-1) + s_{12} + s_{13}\right].
\end{aligned}
\tag{34}
$$

Expressions are rather lengthy in terms of the PaVe functions so that here we only present the expression for the dominant $H_a$ function, which can be written as

$$H_{PV_a}(s_{12}, s_{13}, m_j^2, m_i^2) = \frac{1}{16\pi^2} \frac{N_{H_a}}{D_{H_a}}, \tag{35}$$

with

$$
\begin{aligned}
D_{H_a} = {}& 4\big(m^4 m_{\ell'}^2 - m^2\big[M^2(2m_{\ell'}^2 - s_{12}) + s_{12}(m_{\ell'}^2 + s_{13})\big] + M^4 m_{\ell'}^2 - M^2 s_{12}(m_{\ell'}^2 + s_{13}) \\
& + s_{12}\big(-2s_{13}m_{\ell'}^2 + m_{\ell'}^4 + s_{13}(s_{12} + s_{13})\big)\big),
\end{aligned}
\tag{36}
$$

and

$$
\begin{aligned}
N_{H_a} = {}& \chi_{k_1} C_0(m^2, M^2, s_{12}, m_W^2, m_i^2, m_W^2) + \chi_{k_2} C_0(m_{\ell'}^2, m_{\ell'}^2, s_{12}, m_W^2, m_j^2, m_W^2) \\
& + \chi_{k_3} C_0(M^2, m_{\ell'}^2, m^2 + M^2 + 2m_{\ell'}^2 - s_{12} - s_{13}, m_i^2, m_W^2, m_j^2) \\
& + \chi_{k_4} C_0(m^2, m_{\ell'}^2, m^2 + M^2 + 2m_{\ell'}^2 - s_{12} - s_{13}, m_i^2, m_W^2, m_j^2) \\
& + \chi_{k_5} D_0(m^2, M^2, m_{\ell'}^2, m_{\ell'}^2, s_{12}, m^2 + M^2 + 2m_{\ell'}^2 - s_{12} - s_{13}, m_W^2, m_i^2, m_W^2, m_j^2),
\end{aligned}
\tag{37}
$$

again $\chi_k$ factors are reported in [14].

We can see that although there are additional contributions associated with the $H_k$ functions, with $k = b, c, d, \dots j$; they are expected to be suppressed, as they correspond to higher-dimensional operators, with respect to the $H_a$ function associated with a $(V-A) \times (V-A)$ operator. Therefore, we will concentrate on the $H_a$ function in order to estimate the box diagram contribution. We also have done a numerical cross-check between the expressions for the $H_a$ function given in terms of the Feynman parameters eq. (30) and the PaVe functions eq. (35). In this case, it turns very complicated and far away of the purpose of this work to obtain an analytical expression for the $H_a$ function in eq. (37) making an expansion for the respective scalar PaVe functions, owing to the number of propagators involved and the dependence on two different neutrino masses. However, we can expect a good approximation through our numerical results, as it happens with the penguin contribution.

We estimate the relevant dependence on the neutrino mass for the $H_a$ function taking several points evaluated and fitting the curve for the real and imaginary parts of the $H_a$ function evaluated in terms of the PaVe functions considering fixed values for the $m_i$, $s_{12}$, and $s_{13}$ parameters. We obtained a good fit of the form

$$H_a = \frac{1}{16\pi^2}\left(Q_{H_a} + \frac{m_j^2}{m_W^4} R_{H_a}\right), \tag{38}$$

where $R_{H_a} \approx 1.5 + i0.007$, for all different $\tau \to \ell^- \ell'^- \ell'^+$ channels, whereas $R_{H_a} \approx 1.5$, for the $\mu^- \to e^- e^- e^+$ channel. These numbers were obtained considering that $\Delta m_{ij}^2 = 10^{-3}$ eV$^2$, and representative values for $s_{12}$ and $s_{13}$ within the corresponding phase space.

## 4 Numerical results

In order to evaluate the respective branching fractions for the $L^- \to \ell^- \ell'^- \ell'^+$ decays we considered the state of the art best-fit values of the three neutrino oscillation parameters [10, 11]. Without loss of generality, we assume the $CP$-conserving scenario, and we use the

following values reported for the mixing angles $\sin^2 \theta_{12} = 0.307(13)$, $\sin^2 \theta_{23} = 0.51(4)$, and $\sin^2 \theta_{13} = 0.0210(11)$, whereas the neutrino mass squared differences are taken as $\Delta m^2_{32} = 2.45(5) \times 10^{-3} \text{eV}^2$ and $\Delta m^2_{21} = 7.53(18) \times 10^{-5} \text{eV}^2$ [6]. We also assume a value of $m^2_1 = (0.06)^2 eV^2$.

Our final results, where the dominant penguin and box contributions are considered, are collected in table 2, where they are compared to those obtained using Petcov's results [5] with updated input. Our predictions are smaller owing to the way of the expansion considered and as a consequence of keeping external masses and momenta in our computations.

Table 2: Branching ratios including all contributions (interferences are not neglected), which are obtained using the current knowledge of the PMNS matrix. The last column values correspond to the approximation where external masses and momenta are neglected [5]. Our results are smaller than those by around one (two) orders of magnitude for the $\mu$ ($\tau$) decays.

| Decay channel | Our Result | Ref. [5] |
|---|---|---|
| $\mu^- \to e^- e^+ e^-$ | $7.4 \cdot 10^{-55}$ | $8.5 \cdot 10^{-54}$ |
| $\tau^- \to e^- e^+ e^-$ | $3.2 \cdot 10^{-56}$ | $1.4 \cdot 10^{-54}$ |
| $\tau^- \to \mu^- \mu^+ \mu^-$ | $6.4 \cdot 10^{-55}$ | $3.2 \cdot 10^{-53}$ |
| $\tau^- \to e^- \mu^+ \mu^-$ | $2.1 \cdot 10^{-56}$ | $9.4 \cdot 10^{-55}$ |
| $\tau^- \to \mu^- e^+ e^-$ | $5.2 \cdot 10^{-55}$ | $2.1 \cdot 10^{-53}$ |

# 5  Conclusion

Opposed to the previous calculation reported in ref. [9], we found that all the different amplitudes for the $L^- \to \ell^- \ell'^- \ell'^+$ decays are strongly suppressed (as they are proportional to the neutrino mass squared). In the particular case of the penguin contribution with two neutrino propagators, we highlight that it is crucial to maintain the dependence on the momentum transfer in the Feynman integrals in order to evaluate the amplitude in the physical region for the neutrino masses. This fact avoids the incorrect logarithmic divergent behavior in the amplitude claimed in ref. [9]. As far as the box contribution is concerned, we found that the dominant term comes from $H_a$ function that is associated with a (V-A)×(V-A) operator. The most important result of our analysis is the confirmation (in agreement with ref. [5]) that any future observation of $L^- \to \ell^- \ell'^- \ell'^+$ decays would imply the existence of New Physics.

# Acknowledgements

I really appreciate the collaboration and all the support given by G. López-Castro and P. Roig in this work. I acknowledge financial support from Conacyt through projects FOINS-296-2016 (Fronteras de la Ciencia), and 236394 and 250628 (Ciencia Básica).

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
