# Peer review of "Revisiting the flavor violating decays of the Tau and Mu leptons in the Standard Model with massive neutrinos"

_SciPost Physics Proceedings, doi:SciPost Phys. Proc. 1, 017 (2019)_

## Round 1 · Referee Report · Anonymous (Referee 1) · 2018-12-12

Report
This contribution reports on a recalculation of all the processes of the kind L− → l−l′−l′+ within the Standard Model with massive neutrinos. Main aim of this calculation is to verify the claim in ref. [9] of BR(τ− → μ−l+l−) ≥ 10−14 within the same model. Ref. [9] calculates this process by performing an expansion around the momentum transfer q^2 = 0. This contribution points out that, being q^2 much larger than the neutrino mass squared, the expansion in ref. [9] amounts to evaluating the amplitude outside the physical region.
The main results of this contributions are summarized in table 2, confirming that, due to the smallness of the (m_neutrino / m_W)^4 ratio, all processes of the kind L− → l−l′−l′+ are hopelessly small within the SM, and any measurement would be a clear sign of new physics.
Before accepting the contribution, I would like to ask whether the relevant paper, ref. [23], has meanwhile been published in a journal.
The main results of this contributions are summarized in table 2, confirming that, due to the smallness of the (m_neutrino / m_W)^4 ratio, all processes of the kind L− → l−l′−l′+ are hopelessly small within the SM, and any measurement would be a clear sign of new physics.
Before accepting the contribution, I would like to ask whether the relevant paper, ref. [23], has meanwhile been published in a journal.

Author: Gerardo Hernández-Tomé on 2018-12-13 [id 377]
(in reply to Report 1 on 2018-12-12)Dear referee, thanks for reading our manuscript. Regarding your question, we have recently received a report by EPJC editor asking only for minor style changes. We sent our answer yesterday. We hope our work will be published soon.

---

## Editorial Decision

published